## [Peer Review File · Nature Communications]

REVIEWER COMMENTS

Reviewer #1 (Remarks to the Author):

The authors performed an experimental study of the nature of the CDW instability and transition in the vanadium based Kagome metals. These are exciting systems that are under a tremendous amount of recent interest, and the methods used (DS and IXS) shed light on a different aspect of the transition. As a result, this study is timely and of high impact. The manuscript is well-written, and the results are properly discussed. As a result, I suggest the manuscript for publication after some possible changes in the discussion.

The only change I suggest is addition of a discussion of the complicated nature of the lattice or CDW instabilities in these structures. In discussing the results and the possible scenarios, the third order terms in the hamiltonian (as a function of the M and L phonons, or equivalently, CDW order parameters) is ignored. This might or might not be OK, but it is not obvious if treating this system as a simple system with only quadratic and quartic terms in the lattice hamiltonian is OK. For example, at some point there is a reference to "the harmonic potential well deforms into a double well", which would not be the case in this system since there are 3rd order terms in the Landau free energy expansion.

Another point that should be addressed is the absence of precursors of the M-point charge order above the transition. While it is possible that L point instability is indeed the primary order parameter, symmetry arguments as well as first principles calculations point to the 3rd order coupling which makes L-CDW order necessarily induce the M-CDW order. In other words, in the ground-state static structure, one cannot have L order without the M order. Whether this applies to the fluctuations as well, is not obvious, but it would be very plausible to expect some precursor visible at M, which is induced by what's going on at L. Can the authors address this point, and explain how what they observe (or rather don't observe) at the M point above the transition is consistent with the lattice hamiltonian?

Finally, In these particular systems, there is benefit in explaining what the authors mean by a "3-fold CDW" in detail. While the field used to think that the meaning of this phrase is obvious, I believe it no longer is: Do the authors refer to the star-of-david phase when they say "3-fold CDW", or are the orthorhombic phases where there are displacements with different wavevectors (M and L) are present are also considered as "3-fold"?

Once these points are addressed, the paper is certainly fit for publication in Nat. Comm..

Reviewer #2 (Remarks to the Author):

The kagome metal CsV_3Sb_5 is a highly interesting material system that features intertwined symmetry breaking orders. Using diffuse scattering and inelastic x-ray scattering, the authors studied the evolution of the $2 \times 2 \times 2$ and $2 \times 2 \times 1$ charge density wave (CDW). Their main discovery is that the precursor diffuse intensity is present only at the $2 \times 2 \times 2$ CDW wave vectors. Overall, I think it is a high quality experimental work with important implications on a fast developing field. However, there are few concerns need to be addressed for further considerations in Nature Communications.

1. I'm confused about "Order-disorder Peierls instability". Do the authors indicate "first-order phase transition"? I also don't think the current experimental data strongly support Peierls instability. It may worth to point out that depending on the electronic structure, a first-order structural phase transition can gap the Fermi surface.

2. Regarding the phonon linewidth broadening shown in Fig. 4. First, I think the error bar should be properly described in the caption of the figure. Nevertheless, the indicated error bar might be too small, at least, for the data presented in Fig. 4.

3. The absence of diffuse scattering at $2 \times 2 \times 1$ CDW wavevector is interesting, but also puzzling. It seems to be related to a resonant x-ray work (arXiv: 2202.13530), where the resonant enhancement is only observed at $2 \times 2 \times 2$ CDW wave vector. Fig. 2a shows the diffuse intensity in the ab -plane, suggesting precursor lattice distortions perpendicular to the c -axis. If that's the case, then why the precursor intensity is absent at the $2 \times 2 \times 1$ CDW wavevector?

Reviewer #3 (Remarks to the Author):

D. Subires et, al. investigate the complex nature of the CDW of AV_3Sb_5 ($A = \text{Rb}, \text{Cs}$) via diffuse scattering and inelastic X-ray scattering. An order-disorder type of CDW without phonon softening is revealed in AV_3Sb_5 . A CDW precursor is also indicated via the diffuse scattering.

In fact, for the nature of the CDW in AV3Sb5, a lot of comprehensive work has been carried out. For instance, npj Quantum Materials 7.1 (2022): 1-7, Phys. Rev. Materials 5, L111801(2021). Specifically, the scenario of the order-disorder type of CDW has already been raised by other works (Phys. Rev. Res.4, 023215,2022, Phy. Rev. Mater.5, L111801, 2021). Due to the limited novelty here, the referee finds it hard for a publication in Nature Communications.

Individual response to referee's questions.

REVIEWER COMMENTS

Reviewer #1 (Remarks to the Author):

The authors performed an experimental study of the nature of the CDW instability and transition in the vanadium based Kagome metals. These are exciting systems that are under a tremendous amount of recent interest, and the methods used (DS and IXS) shed light on a different aspect of the transition. As a result, this study is timely and of high impact. The manuscript is well-written, and the results are properly discussed. As a result, I suggest the manuscript for publication after some possible changes in the discussion.

We thank the referee for his/her valuable comments and positive feedback that allowed us to substantially improve the manuscript, especially the Discussion section. In the following, we are giving a detailed response to his/her questions.

The only change I suggest is addition of a discussion of the complicated nature of the lattice or CDW instabilities in these structures. In discussing the results and the possible scenarios, the third order terms in the hamiltonian (as a function of the M and L phonons, or equivalently, CDW order parameters) is ignored. This might or might not be OK, but it is not obvious if treating this system as a simple system with only quadratic and quartic terms in the lattice hamiltonian is OK. For example, at some point there is a reference to "the harmonic potential well deforms into a double well", which would not be the case in this system since there are 3rd order terms in the Landau free energy expansion.

We thank the referee for his/her comment. As detailed in the response to his/her second question, our data suggest that, at $T > T_{CDW}$, the instability at the L point is the leading order parameter and the coupling with the secondary order parameter at M is small or negligible. This leads us to assume that the dynamics of the CDW implies a deformation of the potential well into a double well, as widely reported in literature for first order phase transitions. However, we fully agree with the referee that this can also not be the case because of the complexity of the CDW of these kagome systems, especially at low temperature, where different experiments/theoretical approaches report different types of ordering patterns that imply a coupling of the M and L phonons. Since our data cannot determine the type of deformation of this potential well, and this can adapt more complex shapes than a simple double potential, we have avoided it and the sentence now reads '*In an order-disorder type transformation, the harmonic potential well deforms and the phase transition...*'.

Another point that should be addressed is the absence of precursors of the M-point charge order above the transition. While it is possible that L point instability is indeed the primary order parameter, symmetry arguments as well as first principles calculations point to the 3rd order coupling which makes L-CDW order necessarily induce the M-CDW order. In other words, in the ground-state static structure, one cannot have L order without the M order. Whether this applies to the fluctuations as well, is not obvious, but it would be very plausible to expect some precursor visible at M, which is induced by what's going on at L. Can the authors address this point, and explain how what they observe (or rather don't observe) at the M point above the transition is consistent with the lattice hamiltonian?

This is a very good question and we acknowledge the referee again, as it has allowed us to deliver a new conclusion of high relevance to the physics of kagome systems and update the Discussion section our manuscript. On general grounds, both DFT (phonon instabilities in the calculation) and symmetry analysis say whether a coupling of order parameters associated with special points in the Brillouin zone is possible but cannot say about the strength of the

coupling parameter, γ_{ML} . Therefore, if M-order appears only because of the coupling to L-order, the fluctuations at M-order will be suppressed especially away from T_{CDW} , which is what our data says.

Following a private communication by Rafael Fernandes, the condition for the susceptibility at the M point to diverge is $\chi^2 \sim \gamma_{ML}$, where χ is the correlation length of the L order. The absence of the DS precursor at M (no enhancement of the M susceptibility) could be either because γ_{ML} is small or because the transition is more strongly first-order, in which case χ never really becomes too large. Below T_{CDW} , the effective conjugate field experienced by the M order parameter is $L^2 \gamma_{ML}$, where L is the order parameter of the L order. The key point is that L itself is temperature dependent, so even if γ_{ML} is temperature independent, the mixing with the M order is generally expected to increase as the temperature is lowered. Moreover, because the mixing also depends on how close the system is to a pure M instability, it is possible that, as T is lowered, the instability becomes closer, such that the mixing increases.

In any case, with the experimental data we show in the manuscript, we cannot quantify the strength of the coupled parameter between L and M points. We have updated the Discussion section of the manuscript with the possible reasons for the absence of DS at M and its implication to the low temperature CDW pattern.

Finally, in these particular systems, there is benefit in explaining what the authors mean by a "3-fold CDW" in detail. While the field used to think that the meaning of this phrase is obvious, I believe it no longer is: Do the authors refer to the star-of-david phase when they say "3-fold CDW", or are the orthorhombic phases where there are displacements with different wavevectors (M and L) are present are also considered as "3-fold"?

We thank the referee for pointing this typo out. What we meant here is 3Q (Q_1 , Q_2 and Q_3) CDW (3Q order) that corresponds to a condensation of either M or L phonons. Then, depending on the possible combinations of the $3Q_M$ and $3Q_L$ vectors, several microscopic 3D-CDW structures can be realized. Indeed, according to our high temperature DS data, the alternating Star of David (SoD) and Trihexagonal (TrH) structures (*LLL* distortion) preserving the C_6 (6-fold) rotational symmetry, is the favorable one. Note that models suggesting the coupling of phonons at M and L points of the hexagonal BZ (lowering the six-fold to two-fold symmetry, MLL pattern) are not consistent with our data at high temperature.

Reviewer #2 (Remarks to the Author):

The kagome metal CsV3Sb5 is a highly interesting material system that features intertwined symmetry breaking orders. Using diffuse scattering and inelastic x-ray scattering, the authors studied the evolution of the $2 \times 2 \times 2$ and $2 \times 2 \times 1$ charge density wave (CDW). Their main discovery is that the precursor diffuse intensity is present only at the $2 \times 2 \times 2$ CDW wave vectors. Overall, I think it is a high quality experimental work with important implications on a fast developing field. However, there are few concerns need to be addressed for further considerations in Nature Communications.

We acknowledge the referee for his/her positive evaluation of our work. We also thank him/her for pointing out the anomalous low error bar in figure 4, which we corrected, and his/her questions that allowed us (similarly to referee 1) to improve the quality of our work and the communication of the results. Next, we are answering the questions one by one.

1. I'm confused about "Order-disorder Peierls instability". Do the authors indicate "first-order phase transition"? I also don't think the current experimental data strongly support Peierls instability. It may worth to point out that depending on the electronic structure, a first-order structural phase transition can gap the Fermi surface.

The referee is correct. We have initially opted for 'Peierls instability' as it was theoretically suggested at the beginning that the 2×2 CDW ground state of the kagomé lattice is an inverse Star of David structure with the transition being driven by a Peierls-like nesting-driven instability at wavevectors connecting the M points at the Brillouin zone boundary. As this scenario seems to be not the case, we have modified the title for 'Order-disorder charge density wave instability in the kagome metal $(\text{Cs,Rb})\text{V}_3\text{Sb}_5$ '.

2. Regarding the phonon linewidth broadening shown in Fig. 4. First, I think the error bar should be properly described in the caption of the figure. Nevertheless, the indicated error bar might be too small, at least, for the data presented in Fig. 4.

The referee is correct in his/her comment about the anomalous low error bars in figure 4. This is a consequence of the fitting procedure. The two phonons ω_1 and ω_2 are very close in energy and overlap (according to the DFT calculations there are 6 modes very close in energy at L), thus a reliable fitting that reflects the experimental trend of the linewidth turns out to be a challenge.

The fitting procedure was carried out following the steps:

- First, we fit the IXS spectrum at 105 K and obtain the ω_1 and ω_2 linewidths. From this, we see that ω_1 is resolution limited.
- Looking at the raw data, it is clear that both ω_1 and ω_2 increase their linewidth at 96 K and at higher temperatures as compared with 105 K scan. Trying to fit the linewidth of ω_1 and ω_2 'from scratch' (namely, allowing the code to reach convergence by itself) gives random values in both ω_1 and ω_2 that do not reflect the experimental trend, a T-dependent broadening.
- Therefore, we decided to fix the linewidth of ω_2 to the value obtained at 105 K and fit ω_1 for the rest of the temperatures. The raw data shows that ω_2 broadens less than ω_1 , or its broadening is merely an effect of the larger broadening experienced by ω_1 . By fixing the linewidth of ω_2 during the whole fitting process, its error is zero, and this also gives an anomalously low error for ω_1 .
- To circumvent this issue, now we have continued with the fitting procedure by releasing the linewidth of ω_2 , thus allowing an uncertainty in ω_2 , and finish the fitting until the convergence is reached. We want to point out that the temperature dependence of ω_1 we show here reflects a linewidth trend as observed experimentally and cannot not be interpreted as the intrinsic phonon lifetime of ω_1 and ω_2 .

The figure 4 has been updated with larger error bars and in the suppl. Inf. we include a description of our fitting procedure.

3. The absence of diffuse scattering at 2^*2^*1 CDW wavevector is interesting, but also puzzling. It seems to be related to a resonant x-ray work (arXiv: 2202.13530), where the resonant enhancement is only observed at 2^*2^*2 CDW wave vector. Fig. 2a shows the diffuse intensity in the ab-plane, suggesting precursor lattice distortions perpendicular to the c-axis. If that's the case, then why the precursor intensity is absent at the 2^*2^*1 CDW wavevector?

We thank the referee for this question. Indeed, a similar comment was pointed out by the referee 1 (we would appreciate the referee to go through the response).

On the other hand, the arXiv: 2202.13530 manuscript (published now in Nat. Comm 13, 6348 (2022)) reports a resonant enhancement of the $2 \times 2 \times 2$ CDW and a different T_{CDW} onset for the $2 \times 2 \times 2$ and the $2 \times 2 \times 1$ CDW order at low pressure, supporting a 'conjoined charge density wave'. Whether the resonant enhancement of the $2 \times 2 \times 2$ order is directly related to our DS data at $T > T_{\text{CDW}}$, is unclear to us, but certainly the 'splitting' of the T_{CDW} onsets at M and L might be an indirect indication of L as leading order parameter.

Reviewer #3 (Remarks to the Author):

D. Subires et al. investigate the complex nature of the CDW of AV₃Sb₅ (A = Rb, Cs) via diffuse scattering and inelastic X-ray scattering. An order-disorder type of CDW without phonon softening is revealed in AV₃Sb₅. A CDW precursor is also indicated via the diffuse scattering.

In fact, for the nature of the CDW in AV₃Sb₅, a lot of comprehensive work has been carried out. For instance, npj Quantum Materials 7.1 (2022): 1-7, Phys. Rev. Materials 5, L111801(2021). Specifically, the scenario of the order-disorder type of CDW has already been raised by other works (Phys. Rev. Res.4, 023215,2022, Phy. Rev. Mater.5, L111801, 2021). Due to the limited novelty here, the referee finds it hard for a publication in Nature Communications.

We thank the referee for his/her time to review our manuscript and bring us literature about the nature of the charge density wave in the kagome AV₃Sb₅. We, nevertheless, disagree with the referee that our work lacks novelty and does not deserve to be published in Nature Communications. Our manuscript adds new information to the physics of kagome AV₃Sb₅ that is relevant to understand the enigmatic behavior of its CDW phase:

- We identify a precursor of the CDW at T=150 K by means of diffuse scattering, indicating that the leading order parameter is the Fermi surface instability at the L point. The precursor is absent for the other instabilities, despite the CDWs being present below T_{CDW}. This hasn't been observed/reported previously. This is a very important information, since the charge fluctuations at the L point indicate that: first, the ordering pattern suggested by our measurements consists on an alternating Star of David and trihexagonal pattern (LLL); second, the six-fold symmetry of the lattice is preserved in the fluctuation regime of the CDW phase; third, the coupling parameter of M and L, γ_{ML} , is rather small or negligible, which calls for a proper theoretical description of the coupling of M and L order parameters at low temperature.
- We have observed the same DS pattern for the case of RbV₃Sb₅ at high temperature, suggesting again the same LLL pattern as CsV₃Sb₅ at high temperature.
- Additionally, we show that the charge precursor is a consequence of frozen transversal phonon fluctuations.
- We show that an order-disorder scenario dominates the dynamics above the CDW transition. We further point out that strong electron phonon interaction or non-adiabaticity can be at the origin of this dynamics.
- We detect a phonon broadening below T*~105K due to the growth of CDW domains on approaching T_{CDW} and identify T* as a temperature where charge density domains may start to percolate.

On the other hand, the work by Ratcliff et al, (Phy. Rev. Mater.5, L111801, 2021) and Wulferding et al., (Phys. Rev. Res.4, 023215,2022) speculate about an order-disorder scenario, to describe a broken six-fold symmetry modulation at lower temperature ~60 K, implying a MLL pattern and a transition to a nematic state, respectively. Nevertheless, our data indicates that the order-disorder scenario plays a critical role for the CDW formation at 95 K and we claim that 6-fold symmetry is preserved in the fluctuation regime. This seems to agree with the recent reports of M. Kang et al. (Nature Materials, (2022)) and an LLL+MMM

charge order pattern at low temperature and. This is new information about our work. The papers by Ratcliff and Wulferding are part of the bibliography.

On the other hand, the work *npj Quantum Materials* 7.1 (2022): 1-7, deals with the microscopic origin of the superconductivity in CsV_3Sb_5 by means of muon spin rotation experiments, but not with the origin of the CDW. If the referee refers to the manuscript, *npj Quantum Materials* volume 7, Article number: 30 (2022), the authors study the possible Star of David pattern by nuclear magnetic resonance at low temperature, which seems to be consistent with the type of pattern we infer from high temperature, thus, both works are complementary and add important information to elucidate the enigmatic nature of the CDW phases in AV_3Sb_5 . This manuscript is now part of the bibliography.

Therefore, our manuscript adds critical information about the physics of kagome metals AV_3Sb_5 at $T > T_{\text{CDW}}$ with important implications to understand the enigmatic low temperature CDW phases and, thus, we believe it deserves to be published in a high impact journal as *Nature Communications*.

REVIEWERS' COMMENTS

Reviewer #1 (Remarks to the Author):

The points I raised have been addressed, and as a result, I suggest the manuscript for publication.

Reviewer #2 (Remarks to the Author):

The authors properly addressed my comments and concerns. The new discussion on the M and L instability shed new light on the origin of CDW in AVS system. I, therefore, happy to support its publication in Nature Communications.

Reviewer #3 (Remarks to the Author):

Now I see the significance of this manuscript. This is a nicely written paper and the conclusion is properly discussed, which will be helpful to the community. The referee suggests a publication on Nature Communication.